# Development of a Prediction Model for Short-Term Remission of Patients with Crohn’s Disease Treated with Anti-TNF Drugs

**DOI:** 10.3390/ijms24108695

**Published:** 2023-05-12

**Authors:** Rosario Medina-Medina, Eva Iglesias-Flores, Jose M. Benítez, Sandra Marín-Pedrosa, Isabel Salgueiro-Rodríguez, Clara I. Linares, Sandra González-Rubio, Pilar Soto-Escribano, Beatriz Gros, Manuel L. Rodríguez-Perálvarez, José L. Cabriada, María Chaparro, Javier P. Gisbert, Eduardo Chicano-Gálvez, Ignacio Ortea, Gustavo Ferrín, Valle García-Sánchez, Patricia Aguilar-Melero

**Affiliations:** 1Gastroenterology Unit, Instituto Maimónides de Investigación Biomédica de Córdoba (IMIBIC), Hospital Universitario Reina Sofía, Universidad de Córdoba, 14004 Córdoba, Spain; rmedina@soportevital.es (R.M.-M.); p.aguilar.melero@gmail.com (P.A.-M.); 2Centro de Investigación Biomédica en Red de Enfermedades Hepáticas y Digestivas (CIBERehd), 28029 Madrid, Spain; 3Gastroenterology Unit, Hospital Universitario de Galdakao, 48960 Galdakao, Spain; 4Gastroenterology Unit, Hospital Universitario de La Princesa, Instituto de Investigación Sanitaria Princesa (IIS-IP), Universidad Autónoma de Madrid, 28006 Madrid, Spain; 5Proteomics Unit, Instituto Maimónides de Investigación Biomédica de Córdoba (IMIBIC), Hospital Universitario Reina Sofía, Universidad de Córdoba, 14004 Córdoba, Spain; eduardo.chicano@imibic.org (E.C.-G.);

**Keywords:** inflammatory bowel disease, Crohn’s disease, SWATH proteomics, predictive biomarkers, anti-TNF-a therapy, vinculin

## Abstract

Therapy with anti-tumor necrosis factor (TNF) has dramatically changed the natural history of Crohn’s disease (CD). However, these drugs are not without adverse events, and up to 40% of patients could lose efficacy in the long term. We aimed to identify reliable markers of response to anti-TNF drugs in patients with CD. A consecutive cohort of 113 anti-TNF naive patients with CD was stratified according to clinical response as short-term remission (STR) or non-STR (NSTR) at 12 weeks of treatment. We compared the protein expression profiles of plasma samples in a subset of patients from both groups prior to anti-TNF therapy by SWATH proteomics. We identified 18 differentially expressed proteins (*p* ≤ 0.01, fold change ≥ 2.4) involved in the organization of the cytoskeleton and cell junction, hemostasis/platelet function, carbohydrate metabolism, and immune response as candidate biomarkers of STR. Among them, vinculin was one of the most deregulated proteins (*p* < 0.001), whose differential expression was confirmed by ELISA (*p* = 0.054). In the multivariate analysis, plasma vinculin levels along with basal CD Activity Index, corticosteroids induction, and bowel resection were factors predicting NSTR.

## 1. Introduction

Crohn’s disease (CD) is a chronic inflammatory bowel disease (IBD) that results from an abnormal immune response to enteric microbes in genetically-predisposed individuals and may affect any location of the gastrointestinal tract. As with other immune-mediated disorders, gender-specific differences influence the onset, course, and therapy of IBD. In particular, women suffering from IBD have a worse quality of life than men; because of this, gender medicine could help address gender-specific issues that arise in the management of IBD patients [1]. The classic therapy of CD consists of different combinations of corticosteroids, immunosuppressive agents, and in some cases, surgery. However, new therapeutic approaches have been proposed or developed in recent years. Diet can play a fundamental role in the treatment of CD by modulating metabolic pathways, stimulating gene expression, and modifying the composition of the microbiota. In addition, diet can increase the positive effects of biological therapies by preventing the relapse of the disease after remission [2]. Among novel biologic drugs, tumor necrosis factor alpha (TNF-a) inhibitors (or anti-TNF agents) such as infliximab, adalimumab, and biosimilars have become a paramount therapeutic modality in CD [3,4]. TNF-a is a ubiquitous cytokine whose blood levels significantly increase during the inflammation process occurring in IBD. Anti-TNF agents bind to soluble TNF-a and its transmembrane precursor, thereby blocking the interaction between TNF-a and type 1 and 2 TNF receptors and the subsequent pro-inflammatory cell signaling. The use of anti-TNF agents promotes mucosal healing, avoids the need for steroids, and reduces the need for surgery and hospitalizations, thus, improving quality of life in IBD patients [5]. However, its effectiveness in patients with CD is heterogeneous. Primary non-response to anti-TNF therapy occurs in up to 40% of IBD patients in randomized trials and in 10–20% of patients in real-life series. Secondary loss of response is also frequent in CD patients, with its incidence ranging from 23% to 46% at 12 months after anti-TNF initiation [6,7]. In addition, the chronic use of TNF-a inhibitors is expensive and carries a significant risk of infections, paradox autoimmune events, lymphoproliferative disorders, demyelinating disease, and heart disease [4].

The identification of non-invasive predictors of response to anti-TNF therapy would result in a more efficient prescription of these drugs, thereby optimizing indications and minimizing side effects and costs [8]. This is particularly attractive in CD given the availability of other biological therapies targeting different molecular pathways (e.g., anti-α4β7 integrin, the anti-p40 subunit of interleukin-12, and interleukin-23) [9]. Several studies have aimed to identify reliable biomarkers of response to anti-TNF agents, most of them focused on pharmacogenomics. Genetic variations involving the Fc receptor, apoptosis, TNF signaling, and autophagy have been identified. Other studies have focused on mRNA abundance in peripheral blood and intestinal tissue, microbiota analysis, or on protein markers [8]. However, none of them was sufficiently accurate to be implemented in routine clinical practice.

Proteomics has shown utility in discovering biomarkers related to IBD [10]. However, no study has found definite proteomic predictors of the primary response to anti-TNF agents in CD hitherto [11]. Sequential Window Acquisition of all THeoretical Mass Spectra (SWATH) is one of the most promising approaches in proteomics. It consists of using the information available in fragment ion spectral libraries to mine the complete fragment ion maps generated using a data-independent mass spectrometry acquisition method [12]. The aim of the present study was to identify biomarkers using SWATH proteomics to predict short term clinical remission (STR) after initiating anti-TNF drugs in patients with CD.

## 2. Results

### 2.1. Descriptive Study and Clinical Predictors of Non-Short-Term Remission

One hundred and thirteen patients were clinically analyzed to determine their short-term clinical response to anti-TNF therapy; of these, 85.8% of patients showed STR and 14.2% NSTR. Table 1 shows the baseline features of patients according to their clinical response to therapy. Regarding the anti-TNF agent prescribed, 50.4% were treated with infliximab or biosimilar, and 49.6% received adalimumab without statistical differences in terms of therapeutic response. Younger patients (NSTR: 53.5 (43.0–59.3) years old vs. STR: 39.0 (27.5–50.0) years old, *p* = 0.005) and with a more recent diagnosis of CD (NSTR: 22.0 (2.5–26.0) vs. STR: 4.0 (1.0–13.0), *p* = 0.021), were more likely to have an STR to anti-TNF drugs. Conversely, corticosteroids induction (NSTR: 50.0% vs. STR: 16.5%, *p* = 0.006), previous bowel resection (NSTR: 62.5% vs. STR: 20.6%, *p* = 0.001), and increased CDAI at baseline (NSTR: 235.5 (149.3–310.0) vs. STR: 91.2 (54.9–166.0), *p* = 0.000) were positively associated with NSTR (Table 1).

### 2.2. Proteomic Markers of Short-Term Remission

Three hundred plasma proteins were quantified in proteomic analysis. As potential primary response biomarkers, we identified 18 differentially expressed proteins (*p* ≤ 0.009 and fold change ≥ 2.4): ENOA, VINC, PDZ, and LIM domain protein 1 (PDLI1), Rho GDP-dissociation inhibitor 2 (GDIR2), Zyxin (ZYX), Fructose-bisphosphate aldolase A (ALDOA), moesin (MOES), glutathione S-transferase omega-1 (GSTO-1), alpha-actinin-1 (ACTN1), Transgelin-2 (TAGL2), serum deprivation-response protein/caveolae-associated protein 2 (SDPR), Rab GDP dissociation inhibitor beta (GDIB), cofilin-1 (COF1), protein S100-A4 (S10A4), Ras suppressor protein 1 (RSU1), pleckstrin (PLEK), 14-3-3 protein zeta/delta (1433Z), and triosephosphate isomerase (TPIS) (Table 2). The functional pathway analysis showed that 17 of these 18 identified proteins are regulated by acetylation. In addition, 14 of these potential biomarkers of response to anti-TNF participate in hemostasis/platelet function (VINC, ALDOA, MOES, ACTN1, TAGL2, SDPR, COF1, and PLEK) and/or are involved in the organization of the cytoskeleton and/or cell adhesion (VINC, PDLI1, GDIR2, ZYX, ALDOA, MOES, ACTN1, TAGL2, SDPR, COF1, S10A4, RSU1, PLEK, and 1433Z), 3 of them are related to the inflammatory response (GSTO1, GDIB, and S10A4) and 3 proteins are involved in the glycolytic process (ENOA, ALDOA, and TPIS).

### 2.3. Validation of ENOA and VINC as Potential Markers of Short-Term Remission

ENOA and VINC were selected to perform the validation of proteomic results through ELISA tests due to their higher level of significance (Table 2). According to ELISA, both proteins showed a non-significant increase in plasma samples from STR patients. However, in the case of VINC, the differential expression bordered on statistical significance when NSTR and STR groups were compared (NSTR: 0.8 (0.2–1.2) vs. STR: 1.2 (0.6–2.0), *p* = 0.054) (Table 1 and Figure 1A). Moreover, when patients in the STR group were stratified into patients with sustained remission (SR) and patients with loss of response (LR) after 12 months of therapy, we found statistically significant differences in plasma levels of VINC between the SR (1.3 (0.8–2.1)) and LR groups (0.7 (0.2–1.8)) (*p* = 0.019), and between the SR and NSTR groups (0.8 (0.2–1.2)) (*p* = 0.014) (Figure 1B). These differences were consistent with previous results obtained in the proteomic analysis for VINC, only for SR (492,127.1 ± 219,002.2) and NSTR groups (106,569.3 ± 65,803.9) (*p* = 0.001).

### 2.4. Prognostic Value of ENOA and VINC in CD Patients Undergoing Anti-TNF Therapy

Next, we evaluated the prognostic value of clinical and pathological parameters to predict the patient’s response to anti-TNF treatment. Age (OR: 1.1 (1.0–1.1), *p* = 0.009)), disease duration (OR: 1.1 (1.0–1.2), *p* = 0.006), corticosteroids induction (OR: 5.1 (1.7–15.5), *p* = 0.004) and mainly, bowel resection (OR: 6.4 (2.1–19.8), *p* = 0.001) and basal CDAI score (OR: 1.0 (1.0–1.0), *p* = 0.000), were independent factors of NSTR (Table 1). In a multivariate analysis, corticosteroids induction (OR: 8.6 (1.7–43.5), *p* = 0.009), bowel resection (OR: 10.5 (2.1–52.0), *p* = 0.004), and basal CDAI score (OR: 1.0 (1.0–1.0), *p* = 0.003) were factors predicting NSTR. The inclusion of basal VINC levels, but not ENOA levels, in the analysis increased the predictive capacity of the model (Table 3). A good discriminant capacity of the adjusted model was obtained (AUC (95% CI) = 0.919 (0.862–0.977)) (Table 4 and Figure 2).

## 3. Discussion

The present pilot study illustrated the utility of SWATH proteomics to identify potential plasma biomarkers associated with a differential therapeutic response to anti-TNF agents in CD. In a recent prospective, multicentre, UK-wide study (PANTS study), younger age was associated with an increased likelihood of therapeutic response to infliximab in patients with CD [13]. Moreover, an increased CDAI score at baseline was strongly associated with NSTR. In addition, the earlier prescription of anti-TNF could save bowel resection, which has also been related to NSTR. Aligning with these data, we found that NSTR was more frequent in older patients and with a longer evolution of the disease. Early anti-TNF therapy may improve the natural history of CD, supporting the currently recommended top-down therapeutic approach according to disease severity and patient characteristics. Patients with aggressive behavior of the disease may not benefit from the traditional step-up approach, which could delay an effective suppression of mucosal inflammation, which, in turn, could evolve to increased disease activity and resistance to other drugs [14]. However, the top-down approach, including the use of biological therapy of CD is not without risks, particularly the severe adverse events associated with the widespread use of biologic drugs. Therefore, the identification of non-invasive biomarkers to predict response to anti-TNF agents would allow a more rational and efficient prescription of biological therapies in clinical practice.

The SWATH proteomics identified plasma proteins involved in the organization of the cytoskeleton and cell junction, hemostasis/platelet function, carbohydrate metabolism, and immune response as potential biomarkers of STR. The cytoskeleton is involved in a number of different cellular functions, such as scaffolding, adhesion, migration, and signaling [15]. It exerts key regulatory functions in the maintenance of cellular barriers by cell-to-cell and cell-to-matrix adhesions. Together with the mucosal layer and the cellular immune system, the epithelial cell layer of the gastrointestinal tract forms the first physical barrier against external factors, including an aberrant intestinal microbiome, and allows for the absorption of nutrients and immune detection. Therefore, the integrity of the cytoskeleton is paramount to maintaining the epithelial barrier integrity, and its disruption is thought to play a critical role in the pathogenesis of CD [16]. In immune-mediated inflammatory diseases, pro-inflammatory mediators can induce cytoskeletal rearrangements that cause inflammation-dependent defects in the intestinal barrier’s function [17]. It is not surprising that maintaining the structure of the cytoskeleton was the biological process that accounts for the largest number of proteins differentially expressed between patients with and without therapeutic response to anti-TNF in the present study. Aligning with the role of TNF-a in damaging cell junctions [18], our results show that the response to anti-TNF agents can be affected by proteins involved in these cellular structures. COF1 is a major actin-polymerization factor that controls the opening of tight junctions in the epithelial barrier and participates in the invasion process of the colonic epithelia by the enteric pathogen Shigella [19]. Moreover, increased expression of phospho-COF1 has been associated with the migratory ability of dendritic cells [20], which are the most potent type of antigen-presenting cells and play a central role in the pathogenesis of IBD [21]. 1433Z is a cadherin-binding protein that is involved in cell-to-cell adhesion. Although 1433Z is implicated in the regulation of a large number of signaling pathways, this protein has not been previously associated with CD. VINC [22] is an actin filament-binding protein that, like RSU1 [23], colocalizes at focal adhesions and is involved in cell–matrix adhesion. Decreased expression of VINC has been described in quiescent and inflamed IBD, suggesting that its abundance change is a primary structural feature of the IBD epithelial layer and is not influenced by the inflammatory process. VINC inhibition could imply a decrease in cytoskeletal stiffness and the alteration of the mechanical properties of the epithelial layer, thus increasing the risk of exposure to bacterial invasion [24]. In our study, VINC was the second more differentially expressed protein and one of the five most overexpressed proteins identified by SWATH proteomics. ELISA test partially confirmed this result, reaching statistical significance when SR patients were compared with LR patients within the STR group, or with patients in the NSTR group, in a larger cohort. Thus, VINC analysis at baseline could be useful to identify SR patients and differentiate them from both LR patients and NSTR patients. 

Other proteins related to the actin cytoskeleton that were found to be differentially expressed in our proteomic analysis were GDIR2 [25], TAGL2 [15,26], ZYX [27], PDLI1 [28], SDPR, MOES, PLEK, and ACTN1 [29]. Upregulation of GDIR1 (another member of the Rho GDI family that regulates reorganization of the actin cytoskeleton) in primary intestinal epithelial cells isolated from patients with CD has been associated with the destruction of epithelial homeostasis under chronic intestinal inflammation [30]. SDPR is a protein involved in signal transduction that has been physically associated with PLEK in platelets. ACTN1 and MOES have been proposed to indirectly associate PLEK with actin [29]. This is particularly interesting because all of them have been previously associated with hemostasis/platelet function, probably due to platelets’ cellular physiology, which requires actin polymerization [31]. Indeed, platelet metabolism has been associated with therapeutic response to infliximab in patients with CD [32]. Platelets are potent immune modulators and effectors which directly recognize and kill pathogens, allow to recruit leukocytes at sites of infection and inflammation, and to facilitate their immune activity [33]. Platelets may induce the differentiation of monocytes into dendritic cells and could stimulate the activation of dendritic cells [34,35]. PLEK activation can be detected early after platelet stimulation [36] and has been associated with actin polymerization and related processes such as cytoskeletal reorganization, platelet aggregation, granule secretion, or cell spreading [29,37,38]. SDPR [39] and MOES [40] may participate in different cytoskeletal reorganization processes and platelet biology processes, such as endocytosis, adhesion, locomotion, and signaling. Moreover, MOES activity is associated with lipopolysaccharide-mediated signal transduction in monocytes/macrophages through the induction of TNF-a secretion [40].

S10A4 is a promising therapeutic target in IBD because, similar to other cytoskeleton-related proteins described above, it also amplifies an inflammatory microenvironment [41]. A prospective study with 118 patients identified serum S10A4 protein as a candidate marker to distinguish between IBD and non-IBD population and between CD subgroups [42]. Moreover, some members of the S-100 protein family, such as S10A8 and S10A9, were identified within the top five differentially expressed genes in mucosal biopsy samples from CD patients, able to predict infliximab response with 100% accuracy [43]. The remaining proteins related to the inflammatory response identified in this study were GDIB and GSTO1. The expression of GDIB has been associated with the reduction of inflammation in the tumor microenvironment [44] and the suppression of metastasis [45], being a predictor of relapse-free survival in colorectal cancer [46]. In addition, GDIB is involved in TNF-a-mediated polymorphonuclear neutrophil apoptosis [47]. According to our results, GSTO-1 deficient mice show a more severe inflammatory response and increased escape of bacteria from the colon to the lymphatic system in a preclinical model of IBD [48].

ENOA is a multifunctional enzyme that is involved in many diseases, including inflammatory disorders [49], being identified as the second most differentially expressed protein in human pathologies [50]. In hematopoietic cells, ENOA serves as a receptor of plasminogen [51], which is upregulated in CD patients who show primary non-response to infliximab [52]. Although the ELISA test could not confirm the results obtained for ENOA, other proteins related to glycolysis, such as ALDOA and TPIS, were also identified by proteomics as low-abundance proteins in NSTR patients. In general, IBD patients show strong seroreactivity against glycolytic enzymes. Autoantibodies against ENOA have been identified in ulcerative colitis and CD [53,54]. Therefore, the decrease in glycolytic enzymes in IBD patients could be associated with a higher production of antibodies against enzymes of this pathway, coinciding with more severe disease. Curiously, the cytoskeleton can also impact the glycolytic pathway since full activation of glycolysis requires remodeling of the actin cytoskeleton [55]. 

In combination with well-known clinical predictors, the biomarkers identified in this study could help to better select CD patients who are candidates for anti-TNF therapy. The measurement of plasma levels of VINC added value to the multivariate model when combined with clinical parameters (basal CDAI score, bowel resection, and corticosteroid induction) to predict the response to anti-TNF drugs. In this way, anti-TNF drugs could be prescribed more safely, choosing those patients with an increased likelihood of therapeutic response. Conversely, patients with expected adverse events and/or poor therapeutic response to anti-TNF could be allocated to other biologic agents using different targets, thus allowing for personalized medicine in CD.

## 4. Materials and Methods

### 4.1. Study Design

We conducted a multicentre, observational, and prospective study of patients with CD who started anti-TNF treatment (infliximab, adalimumab, or biosimilar infliximab). The choice of the anti-TNF agent was at the discretion of the responsible physician, and drugs were prescribed according to the licensed dosing schedule. The study protocol was conducted according to the ethical principles contained in the Declaration of Helsinki, and the research protocol was approved by the Andalusian Biomedical Ethics Committee (record number 225, reference 2421). All the patients were informed and provided written consent to participate. 

### 4.2. Study Population

One hundred and thirteen adult patients suffering from CD, naïve to anti-TNF drugs and with indication to anti-TNF therapy due to clinical, endoscopic, or radiological activity, were enrolled at three academic hospitals: Hospital Universitario Reina Sofía (Córdoba, Spain), Hospital Universitario de la Princesa (Madrid, Spain) and Hospital Universitario de Galdakano (Bizkaia, Spain). Therapeutic response was evaluated 12 weeks after the initiation of the anti-TNF drug, and patients were then classified as patients with STR or patients with non-STR (NSTR). The primary outcome of the study was the absence of clinical response, evaluated according to the Crohn’s Disease Activity Index (CDAI) in luminal disease and to the Fistula Drainage Assessment in perianal disease. The criteria to consider the absence of response were: CDAI > 150; endoscopic or radiologic activity; purulent drainage after a gentle finger compression; new fistula; dose escalation; switch to a different drug; addition of other immunosuppressants; surgery or endoscopic dilation [56,57]. STR patients were additionally stratified into patients with SR and patients with LR after 12 months of therapy.

### 4.3. Plasma Samples

Plasma samples were collected within 2 days prior to the first dose of the anti-TNF agent in EDTA-K3 9 mL tubes (K3E VACUETTE^®^, Greiner Bio-One, Kremsmünster, Austria). All samples were spun within the first hour after collection, at 3000 rpm and 4 °C for 10 min. Plasma was immediately recovered and stored at −80 °C until analysis.

### 4.4. Sample Preparation for LC-MS Analysis

The proteomic analysis was performed on a subset of 20 patients randomly selected and classified according to the primary outcome of the study as STR patients (*n* = 16; 8 SR and 8 LR) or NSTR patients (*n* = 4). The 14 most abundant proteins in the blood were depleted using the Hu-14 Multiple Affinity Removal System kit (Agilent Technologies, Wilmington, DE, USA) following the manufacturer’s instructions. Non-depleted proteins were concentrated using 5000 molecular weight cut-off spin concentrators (Agilent Technologies). Samples were then cleaned to remove any contaminant by protein precipitation with TCA/acetone and solubilized in 50 µL of 0.2% RapiGest SF (Waters, Milford, MA, USA) in 50 mM ammonium bicarbonate. The total protein content was measured using the Qubit Protein Assay Kit (Thermo Fisher Scientific, Waltham, MA, USA), and 50 µg of protein was digested with trypsin following a protocol adapted from Vowinckel et al. [58]. Briefly, protein samples were incubated with 5 mM dithiothreitol at 60 °C for 30 min and then with 10 mM iodoacetamide at room temperature and in darkness for 30 min. Sequencing Grade Modified Trypsin (Promega, Madison, WI, USA) was added at a 1:40 trypsin:protein ratio and incubated at 37 °C for 2 h; the same amount of trypsin was again added and incubated at 37 °C for another 15 h. RapiGest was then precipitated by centrifugation after incubating with 0.5% trifluoroacetic acid at 37 °C for 1 h. The final volume was adjusted with milli-Q water and acetonitrile to a final concentration of 0.5 µg peptide/µL, 2.25% acetonitrile, and 0.2% trifluoroacetic acid.

### 4.5. Creation of the Spectral Library

In order to build the MS/MS spectral libraries, the peptide solutions were analyzed by a shotgun data-dependent acquisition (DDA) approach using nano-LC-MS/MS. To obtain a good representation of the peptides and proteins present in all 20 samples, pooled vials were prepared that consisted of equivalent mixtures of the original samples. One μg of each pooled sample (2 μL) was separated into a nano-LC system Ekspert nLC415 (Eksigent, Dublin, CA, USA) using an Acclaim PepMap C18 column (75 μm × 25 cm, 3 µm, 100 Å) (Thermo Fisher Scientific) at a flow rate of 300 nL/min. Water and acetonitrile, both containing 0.1% formic acid, were used as solvents A and B, respectively. The gradient run consisted of 5% to 30% B for 120 min, 10 min at 90% B, and finally 20 min at 5% B for column equilibration, for a total run time of 150 min.

Once the peptides were eluted, they were directly injected into a hybrid quadrupole-TOF mass spectrometer Triple TOF 5600+ (Sciex, Redwood City, CA, USA), operated with a “top 65” data-dependent acquisition system in positive ion mode. A NanoSpray III ESI source (Sciex) was used for the interface between nLC and MS, with a voltage application of 2600 V. The acquisition mode consisted of a 250 ms survey MS scan from 350 to 1250 *m*/*z*, followed by an MS/MS scan from 230 to 1700 *m*/*z* (60 ms acquisition time, rolling collision energy) of the top 65 precursor ions from the survey scan, for a total cycle time of 4.2 s. The fragmented precursors were then added to a dynamic exclusion list for 15 s. Any singly charged ions were excluded from the MS/MS analysis.

Peptide and protein identifications were performed using Protein Pilot software (version 5.0.1, Sciex) with a human Swiss-Prot concatenated target-reverse decoy database (downloaded in March 2016), which contains 20,200 target protein sequences, specifying iodoacetamide as Cys alkylation. The false discovery rate (FDR) was set to 0.01 for both peptides and proteins. The MS/MS spectra of the identified peptides were then used to generate the spectral library for SWATH peak extraction using the add-in for PeakView Software (version 2.1, Sciex) MS/MSALL with SWATH Acquisition MicroApp (version 2.0, Sciex). Peptides with a confidence score above 99% (as obtained from the Protein Pilot database search) were included in the spectral library.

### 4.6. Relative Quantification by SWATH Acquisition

Individual samples were analyzed using a Data-Independent Acquisition method. Each sample (2 μL) was analyzed using the LC-MS equipment and LC gradient described above for building the spectral library instead of using the SWATH-MS acquisition method. The method consisted of repeating a cycle with 60 TOF MS/MS scans (300 to 1600 *m*/*z*, high sensitivity mode, 90 ms acquisition time) of overlapping sequential precursor isolation windows of variable width (1 *m*/*z* overlap) covering the 350 to 1250 *m*/*z* mass range with a previous TOF MS scan (350 to 1250 *m*/*z*, 50 ms acquisition time) for each cycle. The total cycle time was 5.5 s. The width of the 60 variable windows was optimized according to the ion density found in the previous DDA runs using a SWATH variable window calculator worksheet from Sciex.

### 4.7. SWATH Data Analysis

The targeted data extraction of the fragment ion chromatogram traces from the SWATH runs was performed by PeakView (version 2.1) using the MS/MSALL with SWATH Acquisition MicroApp (version 2.0). This application processed the data using the spectral library created from the shotgun data. Up to 10 peptides per protein and 12 fragments per peptide were selected, based on signal intensity; any shared and modified peptides were excluded from the processing. Ten-minute windows and 20 ppm widths were used to extract the ion chromatograms; SWATH quantitation was attempted for all proteins in the ion library that were identified by ProteinPilot with an FDR below 1%. The retention times from the peptides that were selected for each protein were realigned in each run according to 118 endogenous peptides from the Apolipoprotein B-100, eluted along the whole time axis. The extracted ion chromatograms were then generated for each selected fragment ion; the peak areas for the peptides were obtained by summing the peak areas from the corresponding fragment ions. PeakView computed an FDR and a score for each assigned peptide according to the chromatographic and spectra components; only peptides with an FDR below 1% were used for protein quantitation. Protein quantitation was calculated by adding the peak areas of the corresponding peptides. MarkerView (version 1.2.1, Sciex) was used for signal normalization; differential abundance was tested by applying the T-test at the protein level, using SPSS version 25.0 (IBM, Chicago, IL, USA).

### 4.8. Functional Pathways Analysis

Functional pathways from the identified candidate biomarkers were analyzed by DAVID Bioinformatics Resources 6.7 [59,60].

### 4.9. Enzyme-Linked Immunosorbent Assay

An Enzyme-Linked Immunosorbent Assay (ELISA) test was used to quantify the protein concentration of human alpha-enolase (ENOA) (Abnova, Taipei, Taiwan) and vinculin (VINC) (Antibodies-online, Aachen, Germany) in plasma samples from 113 patients (97 STR and 16 NSTR). The assay was carried out following the manufacturer’s instructions, and the obtained data were used to validate the proteomics results.

### 4.10. Statistical Analysis

Categorical variables were displayed in frequency tables, whereas continuous variables were expressed as mean and standard deviation, or as median and interquartile ranges (IQR) for those with asymmetric distribution. The Kolmogorov–Smirnov test was used to check the normal distribution. The Chi-square test was used for frequencies, Student’s T- or ANOVA tests for quantitative variables with normal distribution, and Mann–Whitney’s U or Kruskal–Wallis for asymmetric distributions. A *p*-value of less than 0.05 was considered statistically significant. All analyses were performed by using SPSS version 25.0.

## 5. Conclusions

To our knowledge, this is the first study that applied a SWATH proteomics approach to identify potential biomarkers to predict therapeutic response to biologic agents in CD. Identified plasma proteins were involved in critical hallmarks of the disease. In this sense, depletion of cytoskeletal proteins in CD may trigger disruption of the intestinal epithelial barrier and inflammation, which could translate into more severe disease and resistance to therapy. VINC was a promising biomarker identified in the study, and its differential expression was partially confirmed by ELISA. The combination of plasma VINC levels and clinical data related to corticosteroid induction, bowel resection, and basal CDAI score showed the best predictive value. The validation of candidate biomarkers identified in this proteomic approach in a larger cohort is needed in order to select the appropriate combination of them that, along with clinical features and other inflammatory markers, allow for a more rational and efficient use of anti-TNF agents in clinical practice.

## 6. Patents

A patent application has been submitted to the Spanish Patent and Trademark Office to protect the intellectual property of the results derived from this study (application number P202330350; registration date 5 April 2023).

## Figures and Tables

**Figure 1 ijms-24-08695-f001:**
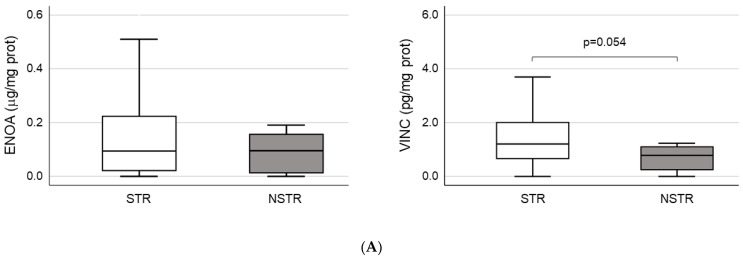
Validation of proteomics results. (**A**) Quantification of plasma concentration of alpha-enolase (ENOA) and vinculin (VINC) by ELISA. The sample size was 97 for the short-term remission (STR) group and 16 for the non-STR (NSTR) group. (**B**) Comparison of plasma levels for VINC by ELISA, between the subgroups of STR patients with sustained remission (SR; *n* = 75), or loss of response (LR; *n* = 22) after 12 months of therapy, and the NSTR group. Statistical significance between groups (*).

**Figure 2 ijms-24-08695-f002:**
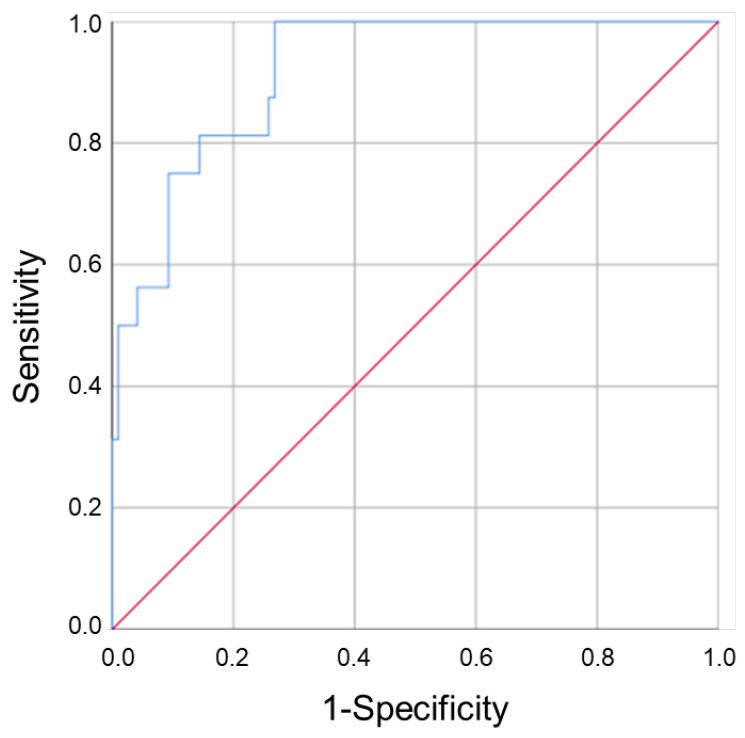
The role of vinculin in the prognostic assessment of CD patient’s response to anti-TNF drugs. The ROC curve showed that the regression model constructed using plasma levels of VINC in combination with clinico-pathological parameters could predict the patient’s response to therapy.

**Table 1 ijms-24-08695-t001:** Relationship between demographic and clinical characteristics and patient response to anti-TNF therapy. Univariate analysis for patients with short-term remission (STR) and patients with non-STR.

	Missing Values	Univariate Analysis			
Variable		NSTR (16)	STR (97)	OR (95% CI)	*p*
Gender (% male)	-	11 (68.8%)	53 (54.6%)	0.5 (0.2–1.7)	0.296
Age (years, median, IQR)	-	53.5 (43.0–59.3)	39.0 (27.5–50.0)	1.1 (1.0–1.1)	0.009 *
Smoking habit (%)	8 (7.1%)				
No		9 (69.2%)	46 (50.0%)	1 (Ref.)	
Yes		4 (30.8%)	32 (34.8%)	0.6 (0.2–2.3)	0.486
Former		0 (0.0%)	14 (15.2%)	0.0 (0.0-)	0.999
Anti-TNF	-				
Infliximab/biosimilar (%)		9 (56.3%)	48 (49.5%)	1 (Ref.)	
Adalimumab (%)		7 (43.8%)	49 (50.5%)	0.8 (0.3–2.2)	0.617
Treatment indication	-				
Luminal disease (%)		15 (93.8%)	84 (86.6%)	1 (Ref.)	
Perianal disease (%)		1 (6.3%)	8 (8.2%)	0.7 (0.1–6.0)	0.745
Both (%)		0 (0.0%)	5 (5.2%)	0.0 (0.0-)	0.999
Immunomodulatory therapy	-				
None (%)		2 (12.5%)	17 (17.5%)	1 (Ref.)	
Azathioprine (%)		13 (81.3%)	74 (76.3%)	1.5 (0.3–7.2)	0.619
Methotrexate (%)		0 (0.0%)	3 (3.1%)	0.0 (0.0-)	0.999
Other (%)		1 (6.3%)	3 (3.1%)	2.8 (0.2–42.0)	0.449
Corticosteroids induction (yes, %)	-	8 (50.0%)	16 (16.5%)	5.1 (1.7–15.5)	0.004 *
Basal BMI (Kg/m^2^) (median, IQR)	10 (8.8%)	23.0 (20.8–24.9)	22.4 (20.5–26.0)	1.0 (0.8–1.1)	0.838
Disease duration (years, median, IQR)	8 (7.1%)	22.0 (2.5–26.0)	4.0 (1.0–13.0)	1.1 (1.0–1.2)	0.006 *
Montreal (age at diagnosis)	11 (9.7%)				
<17 (%)		0 (0.0%)	7 (7.8%)	1 (Ref.)	
17–40 (%)		9 (75.0%)	64 (71.1%)	2.3 × 10^8^ (0.0-)	0.999
>40 (%)		3 (25.0%)	19 (21.1%)	2.6 × 10^8^ (0.0-)	0.999
Montreal location	3 (2.7%)				
Ileal (%)		8 (53.3%)	39 (41.1%)	1 (Ref.)	
Colonic (%)		1 (6.7%)	16 (16.8%)	0.3 (0.0–2.6)	0.281
Ileocolonic (%)		6 (40.0%)	37 (38.9%)	0.8 (0.3–2.5)	0.689
Isolated upper disease (%)		0 (0.0%)	3 (3.2%)	0.0 (0.0-)	0.999
Behavior	8 (7.1%)				
Inflammatory (%)		5 (38.5%)	56 (60.9%)	1 (Ref.)	
Stricturing (%)		1 (7.7%)	16 (17.4%)	0.7 (0.1–6.4)	0.753
Fistulizing (%)		7 (53.8%)	18 (19.6%)	4.4 (1.2–15.4)	0.023 *
Fistulizing and Stricturing (%)		0 (0.0%)	2 (2.2%)	0.0 (0.0-)	0.999
Perianal disease (yes, %)	1 (0.9%)	2 (12.5%)	32 (33.3%)	0.3 (0.1–1.3)	0.111
Extraintestinal manifestation (yes, %)	8 (7.1%)	2 (15.4%)	14 (15.2%)	1.0 (0.2–5.1)	0.987
Appendicectomy (yes, %)	8 (7.1%)	1 (7.7%)	8 (8.7%)	0.9 (0.1–7.6)	0.904
Bowel resection (yes, %)	-	10 (62.5%)	20 (20.6%)	6.4 (2.1–19.8)	0.001 *
Perianal surgery (yes, %)	-	0 (0.0%)	18 (18.6%)	0.0 (0.0-)	0.998
Basal CDAI score (AU, median, IQR)	-	235.5 (149.3–310.0)	91.2 (54.9–166.0)	1.0 (1.0–1.0)	0.000 *
Basal hemoglobin (g/L, mean ± SD)	9 (8.0%)	12.7 ± 1.0	13.0 ± 1.5	0.8 (0.6–1.3)	0.384
Basal WBC (10^3^/μL, median, IQR)	9 (8.0%)	7.5 (5.5–11.7)	7.6 (5.6–10.4)	1.1 (0.9–1.2)	0.561
Basal platelets (10^3^/μL, median, IQR)	10 (8.8%)	304.0 (223.0–356.5)	312.5 (263.5–363.5)	1.0 (1.0–1.0)	0.235
Basal albumin (g/dL, mean ± SD)	22 (19.5%)	4.0 ± 0.4	4.0 ± 0.5	0.9 (0.2–3.8)	0.940
Basal ferritin (ng/mL, median, IQR)	19 (16.8%)	119.5 (51.2–187.3)	55.0 (26.0–114.9)	1.0 (1.0–1.0)	0.449
Basal CRP (mg/L, median, IQR)	14 (12.4%)	2.3 (0.6–3.4)	4.4 (0.6–14.6)	1.0 (0.9–1.0)	0.253
Basal ESR (mm/h, median, IQR)	34 (30.1%)	35.0 (10.8–85.8)	27.0 (11.0–37.0)	1.0 (1.0–1.0)	0.121
Basal ENOA (μg/mg protein)	1 (0.9%)	0.1 (0.0–0.2)	0.1 (0.0–0.2)	0.1 (0.0–4.7)	0.199
Basal VINC (pg/mg protein)	-	0.8 (0.2–1.2)	1.2 (0.6–2.0)	0.7 (0.4–1.2)	0.171

* BMI, body mass index; CDAI, Crohn’s Disease Activity Index; WBC, white blood cells; CRP, C-reactive protein; ESR, erythrocyte sedimentation rate; ENOA, alpha-enolase; VINC, vinculin; OR, odds ratio; CI, confidence interval.

**Table 2 ijms-24-08695-t002:** Differentially expressed proteins identified in plasma samples of Crohn’s disease patients with STR to anti-TNF. Statistical significance by t-test (*p* ≤ 0.01).

Protein ID	Protein	*p*	Fold Change	Biological Process	Molecular Function
P06733	ENOA	0.0001	3.9	Glycolysis/Plasminogen activation/Transcription regulation	DNA binding/Lyase/Repressor
P18206	VINC	0.0007	4.6	Cell adhesion/Cytoskeleton; Hemostasis/Platelet function	Actin binding
O00151	PDLI1	0.0013	2.4	Cell adhesion/Cytoskeleton	Actin binding
P52566	GDIR2	0.0013	4.9	Cell adhesion/Cytoskeleton	GTPase activity
Q15942	ZYX	0.0014	5.5	Cell adhesion/Cytoskeleton	Metal binding/RNA binding
P04075	ALDOA	0.0021	3.3	Glycolysis; Cell adhesion/Cytoskeleton; Hemostasis/Platelet function	Actin binding/Fructose-bisphosphate aldolase activity
P26038	MOES	0.0023	2.5	Cell adhesion/Cytoskeleton; Hemostasis/Platelet function	Actin binding
P78417	GSTO1	0.0025	3.2	Inflammatory response	Oxidoreductase/Transferase
P12814	ACTN1	0.0025	3.2	Cell adhesion/Cytoskeleton; Hemostasis/Platelet function	Actin binding
P37802	TAGL2	0.0027	6.4	Cell adhesion/Cytoskeleton; Hemostasis/Platelet function	Cadherin binding
O95810	SDPR	0.0027	4.2	Cell adhesion/Cytoskeleton; Hemostasis/Platelet function	Lipid binding
P50395	GDIB	0.0028	3.0	Inflammatory response	GTPase activation
P23528	COF1	0.0029	3.0	Cell adhesion/Cytoskeleton; Hemostasis/Platelet function	Actin binding
P26447	S10A4	0.0031	3.3	Cell adhesion/Cytoskeleton; Inflammatory response	Actin binding
Q15404	RSU1	0.0037	3.9	Cell adhesion/Cytoskeleton	Positive regulation of GTPase activity
P08567	PLEK	0.0048	5.7	Cell adhesion/Cytoskeleton; Hemostasis/Platelet function	Protein binding
P63104	1433Z	0.0078	4.2	Cell adhesion/Cytoskeleton	Protein binding
P60174	TPIS	0.0093	3.6	Glycolysis	Isomerase/Lyase

**Table 3 ijms-24-08695-t003:** Multivariate analysis to predict NSTR to anti-TNF therapy in Crohn’s disease. The analysis was performed separately, including exclusively clinical variables (without -w/o- VINC or ENOA), and including clinical variables together with (w/) ENOA or VINC.

	Multivariate Analysis					
	w/o VINC/ENOA		w/ ENOA		w/ VINC	
Variable	OR (95% CI)	*p*	OR (95% CI)	*p*	OR (95% CI)	*p*
Age (years)	-	-	x	x	x	x
Corticosteroids induction (%)	8.6 (1.7–43.5)	0.009 *	12.8 (2.4–68.8)	0.003 *	14.3 (2.6–77.7)	0.002 *
Disease duration (years)	-	-	x	x	x	x
Bowel Resection (%)	10.5 (2.1–52.0)	0.004 *	13.2 (2.5–68.9)	0.002 *	14.8 (2.8–78.0)	0.001 *
Basal CDAI score (AU)	1.0 (1.0–1.0)	0.003 *	1.0 (1.0–1.0)	0.001 *	1.0 (1.0–1.0)	0.002 *
Basal ENOA (µg/mg protein)	x	x	0.0 (0.0–1.7)	0.067	x	x
Basal VINC (pg/mg protein)	x	x	x	x	0.5 (0.3–0.9)	0.032 *

(*) Statistically significant as a predictor in the multivariate analysis.

**Table 4 ijms-24-08695-t004:** Area under the receiver operating characteristic curve (AUROC) of the multivariate models in Figure 2. The table shows the comparison of AUROC obtained for the single parameters included in the adjusted model and for their combination w/ or w/o vinculin.

Variable	AUROC	*p*	95% CI
Basal VINC (pg/mg)	0.651	0.054	(0.500–0.802)
Corticosteroids induction	0.668	0.032	(0.511–0.824)
Bowel resection	0.709	0.007	(0.562–0.857)
CDAI score	0.829	0.000	(0.718–0.940)
Adjusted model (w/o VINC)	0.904	0.000	(0.838–0.970)
Adjusted model (w/ VINC)	0.919	0.000	(0.862–0.977)

## Data Availability

The data presented in this study are available on request from the corresponding author. The data are not publicly available due to a patent application.

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
