# Peer review of "Development of a Prediction Model for Short-Term Remission of Patients with Crohn’s Disease Treated with Anti-TNF Drugs"

_ijms, 2023, doi:10.3390/ijms24108695_

Round 1
Reviewer 1 Report
In the manuscript by Medina-Medina et al, the authors aimed to identify reliable markers of response to anti-tumor necrosis factor (TNF) drugs in patients with Crohn's disease (CD). They stratified a cohort of 113 anti-TNF naive patients with CD according to clinical response as short term remission (STR) or non-STR (NSTR) at 12 weeks of treatment. By utilization of SWATH proteomics to identify potential plasma biomarkers associated with a differential therapeutic response to anti-TNF agents in CD, they compared protein expression profiles of plasma samples in a subset of patients from both groups prior to anti-TNF therapy, and identified 18 differentially expressed proteins as candidate biomarkers of STR, among them, vinculin was one of the most deregulated proteins whose differential expression was confirmed by ELISA. The measurement of plasma levels of vinculin added value to the multivariate model when combined with clinical parameters (basal CD activity index score, bowel resection, and corticosteroids induction) to predict response to anti-TNF drugs.
The validation of candidate biomarkers identified in this proteomic approach in a larger cohort would help with selection of the appropriate combination of candidate biomarkers that, along with clinical features and other inflammatory markers, allow for a more rational and efficient use of anti-TNF agents in clinical practice.
The study was very interesting with clinically significant findings. The manuscript was very well written.
Reviewer 2 Report
Dear Authors,
This is a good piece of work, fascinating. I have only a suggestion.
I would add, in the Introduction, that also diet therapy is becoming a tool to improve Crohn's Disease (Caio G, Lungaro L, Caputo F, Zoli E, Giancola F, Chiarioni G, De Giorgio R, Zoli G. Nutritional Treatment in Crohn's Disease. Nutrients. 2021 May 12;13(5):1628. doi: 10.3390/nu13051628. PMID: 34066229; PMCID: PMC8151495.).
I would highlight this, as diet is very often neglected whereas it is very important to maintain remission.
It is also helpful to explain to the reader that nowaday, the gender-tailored approach is becoming of great use when treating patients, especially females. (Lungaro L, Costanzini A, Manza F, Barbalinardo M, Gentili D, Guarino M, Caputo F, Zoli G, De Giorgio R, Caio G. Impact of Female Gender in Inflammatory Bowel Diseases: A Narrative Review. J Pers Med. 2023 Jan 17;13(2):165. doi: 10.3390/jpm13020165. PMID: 36836400; PMCID: PMC9958616.)
Best regards.
Reviewer 3 Report
It is a very good paper in the field of proteomics.
I have the following suggestions:
- To revise the phrase on lines 119-120. Unfinished sentence.
- To revise the information from the phrase that begins on row 120. In my opinion, the results from the ST and LR subgroups should be interpreted comparatively, and only after that I think it should be compared with those obtained for the NSTR group.
- The definition of the abbreviations SR and LR in row 294 must be moved to rows 121 and 122.
- To include references in the Materials and methods section.
- To introduce the results obtained for ENOA, by using the multivariant analysis. The proteins with the highest significance (ENOA and VINC) were selected for validating the results by the ELISA technique. It has been shown that the SWAT technique improves the knowledge of physiological and pathological mechanisms, but also for applied and translational research, by allowing the identification of biomarkers expression changes for which classical biochemical analysis in serum is irrelevant. For the two proteins, the result was also validated in patients treated with anti-TNF to prove that introducing VINC as a prognostic factor increases the predictive capacity of the multivariant analysis model. It is also interesting to see the conclusion of ENOA expression change.
Round 2
Reviewer 3 Report
Now, in my opinion, the manuscript can be accepted in this revised form.